# Mobile Apps for Helping Patient-Users: Is It Still Far-Fetched?

**Charalampos Siristatidis** [1,*] **, Abraham Pouliakis** [2] **, Vasilios Karageorgiou** [3] **and Paraskevi Vogiatzi** [1]

1   3rd Department of Obstetrics and Gynecology, Assisted Reproduction Unit, "Attikon" University Hospital, Medical School, National and Kapodistrian, University of Athens, 12462 Athens, Greece; evivogiatzi@gmail.com

2   2nd Department of Pathology, "Attikon" University Hospital, National and Kapodistrian University of Athens, 12462 Athens, Greece; apou1967@gmail.com

3   2nd Department of Psychiatry, "Attikon" University Hospital, National and Kapodistrian University of Athens, 12462 Athens, Greece; vaskarageorg@hotmail.com

*   Correspondence: csyristat@med.uoa.gr; Tel.: +30-6932294994

**Abstract:** Emergence of health-related smartphone applications and their wide dissemination in public as well as healthcare practitioners has undergone criticism under the scope of public health. Still, despite methodological issues curbing the initial enthusiasm, availability, safety and, in certain cases, documented efficacy of these measures has secured regulatory approval. Bearing in mind these pitfalls, we describe the necessary steps towards implementation of deep learning techniques in the specific clinical context of women's health and infertility in particular.

**Keywords:** mobile health; infertility; neural networks; smartphone applications

## 1. Introduction

We have come to an era where health information is readily available. Smartphone utilities demonstrate rapid advancement and the market offers applications (apps) covering almost every aspect of human life, accessible anywhere and anytime; these have engaged practitioners, providing handy tools for information access, monitoring, telemedicine and even supporting medical decisions [1–7].

Medical apps with ad hoc devices are increasingly used by professionals. In certain conditions, e.g., type 1 & 2 diabetes [1,2], they have been proven effective and are currently recommended. These benefits also apply to patients/users, where interventions of notifying them have promoted disease monitoring, management, and health education. Still, enthusiasm for novelty and/or financial interest may leave misleading or inaccurate information unnoticed, ultimately risking users' personal welfare.

Currently, mobile applications are even targeting consumers starting from a very young age, and a lot of best-selling apps for education are focusing on children [3]. Meanwhile, there is a positive attitude of parents on the use of mobile technology by their children, even from a very young age; and this renders the high the quality (and quantity) of apps a "must" [4].

In the medical context, utilization of Artificial Intelligence (AI) could provide new levels of information provisioning [8] and management for various medical modalities, with continuous learning and self-adapting characteristics. One of the enabling approaches of AI is Artificial Neural Networks (ANNs): these systems have the capability to learn and optimize their response by learning from the same problem thousands of times and then by adjusting their response according to the feedback they receive [9].

Methodological weaknesses have been pointed out in apps targeting frequent clinical problems. Examples include: data insufficiency for algorithm training [10], varying comprehensiveness and quality of apps and lack of transparency [11], as in the case of apps that target cancer patients [12]. Severe concerns for sample representativeness are raised in image-analysis-based melanoma diagnosis [13]. Few apps provide comprehensive information on all aspects [14], including effectiveness, side effects, and contraindications. Another issue is the discrepancy between expected and real-world user behaviors [15].

In this paper, we briefly describe the methodological weaknesses of apps targeting frequent clinical problems through a comprehensive literature search, and based on these, we propose specific key approaches integrating AI into mobile apps in the specific clinical context of improving patient's health.

## 2. Methodology

Bearing in mind the aforementioned methodological weaknesses related to apps that target clinical problems in mind, we performed a relevant literature search. The key words selected were the following: "mobile application; health; mobile apps; self-management of chronic disease; digital divide; health-promoting behavior; mHealth; smartphone mHealth; mobile phone; self-management; systematic review; telemedicine; humans; health behavior; multimedia; text messaging; cell phone; health status; patient satisfaction; prenatal care/methods; primary prevention/methods; data accuracy; minicomputers; artificial intelligence; artificial neural network; ANN".

Based on our findings, and critically evaluating the main points of our search, we sought to (a) narratively synthesize the most relevant findings and (b) construct a schematic framework that could lead to the improvement of the role of a mobile app to users for their health tracking and control.

## 3. Results

### 3.1. Our Proposal

A schematic representation of a workflow that could be applied for the construction of an app that incorporate an ANN is given in Figure 1.

### 3.2. The First Phase: Evaluation and Appraisal

Prior to launch, akin to medical literature, apps advertised as health-related should undergo peer review [16]. Another suggestion is to collect feedback from both professionals and regular users, for software developers to extrapolate, exchange opinion and suggest improvements. Apps' sources should be always examined in terms of scientific origin, while Health Insurance accountability should be always considered, in terms of privacy of health information, security of electronic records, administrative simplification, and insurance portability [17].

Rigorous evaluation [18], validation and best-practice standards for medical apps are greatly needed to ensure a fundamental level of quality and safety. It is important for policy-makers to enforce dialogue with users to understand their needs [19], and maximize benefits.

Regulatory assemblies, such as the FDA [20], are required to monitor and supervise the process of app publication, marketing and implementation, in order to identify [5], review and determine accuracy and potential health hazards on their use. A recent PLoS One editorial [21] underlined the further limitation of code peer-review, a timely endeavor, and the need for post-publication review. A centralized approach of fewer apps, with multiple inputs from different sources, abiding to the open source culture [22], with collaboration of medical specialists and developers could prove beneficial.

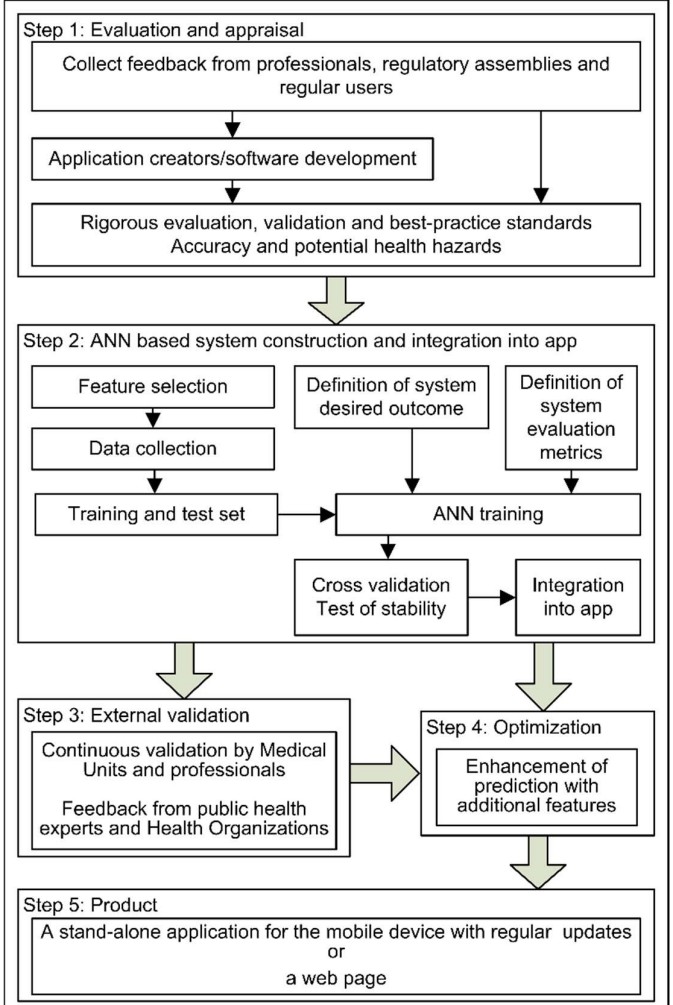

**Figure 1.** Workflow of the proposed approach, from the conception and appraisal of the app towards the creation/integration and subsequent maintenance and enhancement of the product.

### 3.3. The Second Phase: Construction of an Ann and Development of an App

Key approaches include the construction of the ANN and the development of the app Together with the use of significant features contributing to the primary clinical outcome. The necessary steps have been described elsewhere [8]. Briefly:

The first step comprises the collection of data, the size of which is heavily related to the nature of the problem, along with the feature extraction and the selection of algorithms in order to create the appropriate input data set;

The second step includes data pre-processing, clearing any inconsistent values and by performing mathematical transformations, along with the separation of the available data into training and test sets;

The third step involves the selection of the appropriate ANN model and the definition of its parameters and characteristics;

The final steps include the evaluation of the system performance using either the test set or an external data set, according to the required and expected performance. Ideally, new participant samples, gathered under different procedures yet comparable to the original data, are used for accuracy assessment (external validation). Satisfactory output of these steps will indicate whether the system has acceptable performance so as to enter a production environment for routine usage. If not, all steps are to be re-evaluated from the beginning. It is also possible to allow the algorithm to update automatically with the registration and input of new participant records.

In parallel to the ANN construction phase, there is the standard phase for the mobile app development, that will present the ANN functionality to the users, through the creation of a user interface.

### 3.4. The Third Phase: External Validation of the New System (App+ANN)

This phase is necessary for the identification of issues that may arise from the use of the new system in a large scale, and includes:

1.  Continuous external validation by other Medical Units and professionals in the field, therefore allowing continuous re-adjustment of efficiency and stability of the ANN. This stage involves an extended review and certification process, achieved through the development and use of standardized vocabularies and interfaces for data exchange, storage and reporting.
2.  Running alongside, a feedback from public health experts and Health Organizations is necessary; this can be accomplished by taking into account factors as: comparative impact of false-positives and false-negatives, model accuracy in independent validation sets (and not the reported test set).

### 3.5. Fourth Phase: Optimization/Enhancements

Further enhancement of predictors with laboratory results, e.g., serum hormonal profile, probably connected in a second ANN, to improve diagnostic accuracy and differentiate participants and their risks according to more elaborate population variations.

### 3.6. Fifth Phase: Product

The final product (app+ANN) can be presented to the end-users (patients) either as a stand-alone application with timely updates according to continuous improvement through novel results from external validation or as a public web page, for example by interfacing a web based system with the ANN [23].

In addition, the new system has to be tested through adversarial attacks [24] for potential weaknesses, in order to ensure that personal data are not exposed, especially data related to medical records and that the model is not biased by abnormal patterns or unexpected anomalies.

## 4. Discussion

Technology for mobile devices and AI is rapidly advancing [6,7]; there is an urgent demand, justified and led by the prevailing patients' needs, to combine the two fields for a controlled development of quality apps [25], taken the weaknesses reported so far into account [7]. In this brief report, we propose specific key approaches intergrading AI into mobile apps, taking into consideration the current knowledge on the existing flaws –both in diagnosis and management proposals- of the latter.

Current technological advancements, especially those related to large interactive touch displays [26], have influenced both educational processes and learning environments, along with patient care concepts. Both the high mobility and the high resolution of offline and online media, as well as the low cost and the capability of interaction has resulted in a rising trend favoring the use of mobile devices from the young population [26]. As such, mobile learning as "the process of coming to know through conversations across multiple contexts amongst people and personal interactive technologies" [27] has been used to cover current needs. Also, new notions have been emerged, such as Micro-learning as an educational teaching method used to train users on multiple platforms [28]. Concurrently, smart phone display screen with various user graphical interfaces have been reported and patented [29,30].

Additionally, digital health constitutes a combination of both software and hardware technology with health care delivery and management, and encompasses a number of domains, from wearable devices to AI, utilizing varied methods of data collection and information flow, including monitoring of chronic conditions, such as respiratory, weight management, metabolic, and cardiovascular

diseases [31], training of medical professionals [32–34] and core uses, such as apps in medical laboratories [35,36]. Another fine example is on cancer care, from its screening and management strategies to survivorship [37] and in personalized medicine, supporting fitness, health education, symptom tracking, and collaborative disease management care coordination [38–40], as well as, apps targeting from aging population [41,42] to children [43].

Of note, there are various published approaches on ANNs and prediction models, in terms of methodological design, outcomes, completeness, efficiency and stability of the system. In our previous work on ANN and prediction of assisted reproduction outcome [8], we have referenced the first ANN published in 1997, providing an overall accuracy of 59% in predicting clinical pregnancy with PPV of 39% and NPV of 82%, whereas other reports thereafter demonstrated predictive accuracy varying between 72 and 82% [44].

In this paper we propose the construction of a synergy of AI / ANNs and mobile apps for optimally personalized strategies in both diagnostic and therapeutic field. ANNs are capable of recalling and evaluating a vast amount of information in a rapid and automated manner providing an objective indication on the outcome of a specific pathological condition, either in diagnosis or treatment, which is utterly consistent with the nature of input parameters and the "model cases" explored. Such steps can include the initial use of features proven so far for their efficacy to positively contribute to the primary outcome; the continuous improvement through feedback from experts, Health Organizations and the users themselves, the integration of laboratory results and their repeated testing for potential weaknesses. Only collaborative, large datasets can provide the necessary sample size required for these approaches to prove their competitive edge over standard methods, such as probabilistic measures. As mentioned before in one of our papers: "Issues to be stressed include the objectivity of the network expert and medical staff responsible for the system "training" as personal evaluation or misinterpretation of symptoms or indications may, for example, falsify data input that could overall affect the performance of the system and its predictive accuracy, the input parameters or the input of "true" or resolved cases during training leaving out any cases where diagnosis or other parameters are undefined or questionable" [8]. Undoubtfully, careful interpretation of the dynamics and potential of such combination (ANN and app) is required and this should be in a harmonic balance with clinical judgment and decision-making. Also, further expansion of the system by incorporating data from research collaborations to include cases with an adverse outcome or other more complex pathological states of disease, could prove beneficial for the research community or at routine clinical implementation [8,37].

In conclusion, we advocate further integration of AI-inspired approaches "behind the clinical scenes", which, to our current knowledge, have been rarely integrated in apps. This could prove extremely useful in emergency cases where healthcare professionals require evidence-based supporting medical opinion in their approach and in circumstances not allowing advisory from other professionals, as also for users themselves to be alerted to seek immediate medical attention in certain pathologies or in certain symptoms development.

**Author Contributions:** Conceptualization, C.S.; methodology, C.S. and A.P.; software, A.P.; validation, V.K. and P.V.; formal analysis, C.S. and A.P.; investigation, C.S. and A.P.; resources, V.K. and P.V.; data curation, V.K. and P.V.; writing—original draft preparation, C.C.; writing—review and editing, C.S., A.P., V.K. and P.V.; visualization, C.S., A.P., V.K. and P.V.; supervision, C.S., A.P., V.K. and P.V.; project administration, C.S., A.P., and P.V.; funding acquisition, C.S. and P.V. All authors have read and agreed to the published version of the manuscript.

**Funding:** This research received no external funding.

**Conflicts of Interest:** The authors declare no conflict of interest.

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
