# Peer review of "Mobile Apps for Helping Patient-Users: Is It Still Far-Fetched?"

_sustainability, doi:10.3390/su12010106_

Round 1

Reviewer 1 Report

The authors presented a manuscript focused on the necessary steps towards implementation of deep learning techniques in the specific clinical context of women’s health and infertility.

The manuscript does not provide any contribution, evaluation or scientific analysis. It is just a quick discussion about the context and the state of the art.

Author Response

Editor’s comments:

Comment 1: The literature review should be more carefully synthesised and structured. The use of sub-headings and signposting would help the reader to follow the argument being developed through the paper.

Authors’ actions: We thank the editor for the comment; we have added more references and more sub-headings in order to provide a better structured document.

Comment 2: There does not appear to be an explicit theoretical framework. Currently the manuscript appears to be somewhat descriptive and a theoretical.

Authors’ actions: We thank the editor for the comment; we have amended the paper accordingly: we have enriched most parts, added a methodology section and new paragraphs on the construction of an ANN and its cooperation for the creation of the app.

Comment 3: The results section requires far greater organisation and structuring. The analysis is too general, and the reported results are somewhat selective. This section needs to be more carefully and systematically constructed.

Authors’ actions: We tried to follow the suggestions, the results section now is structured more systematically and augmented by a figure to facilitate the reader to follow the concept.

Comment 4: Further, the analysis and findings must be critical and interpretive rather than just descriptive.

Authors’ actions: We thank the editor for the comment; we tried to conform with the suggestions.

Comment 5: The final discussion and conclusion should make it clear how the findings contribute to new knowledge.

Authors’ actions: We thank the editor for the comment; we tried to conform with the suggestions.

Comment 6: The Methodology lacked suitable detail.

Authors’ actions: We thank the editor for the comment; we have added a separate section.

Comment 7: Research methodology based on research goals is poor on absent.

Authors’ actions: We thank the editor for the comment; we tried to conform with the suggestions.

Comment 8: More recent bibliography is necessary. Furthermore, the reference list is a little bit weak. Before I can make a final decision on the paper, please refer to more references and upload a new version. It is suggested that the author(s) can consider the following papers related to mobile learning usage, etc. to strengthen the background of the study: • Papadakis, St., & Kalogiannakis, M. (2017). Mobile educational applications for children. What educators and parents need to know. International Journal of Mobile Learning and Organisation (Special Issue on Mobile Learning Applications and Strategies), 11(3), 256-277. • Papadakis, S., Kalogiannakis, M., & Zaranis, N. (2019). Parental involvement and attitudes towards young Greek children’s mobile usage. International Journal of Child-Computer Interaction. doi: https://doi.org/10.1016/j.ijcci.2019.100144. • Papadakis, St., & Kalogiannakis, M. (2020). A Research Synthesis of the Educational Value of Self-proclaimed Mobile Educational Applications for Young Age Children. In S. Papadakis, & M. Kalogiannakis (Eds.), Mobile Learning Applications in Early Childhood Education (pp. 1-19). Hershey, PA: IGI Global. doi:10.4018/978-1-7998-1486-3.ch001

Authors’ actions: We have added a new paragraph in the introduction highlighting the role of mobile apps, especially for the future consumers of health apps (i.e. nowadays children). Moreover the role of modern technology is highlighted with more emphasis in the discussion. We have also added the proposed references.

Comment 9: The discussion should be more concise and the outcomes should be discussed in relation to the existing research.

Authors’ actions: We thank the editor for the comment; we tried to conform with the suggestions.

Comment 10: Recommendations should also be given for practice and further research.

Authors’ actions: We thank the editor for the comment; we have added this in the discussion section.

Comment 11: Similarity check with iThenticate revealed a similarity index of 6% which is considered appropriate.

Authors’ actions: Thank you for pointing, no action.

Reviewer 2 Report

Comments to the Author
Comment: I had the privilege of reviewing the manuscript titled: “Mobile apps for helping patient-users: is it still farfetched?” by Dr. Siristatidis and colleagues.  The manuscript is well written and adds to the field of study. 

Comment 1: provide references to this statement in the introduction “Smartphone utilities demonstrate rapid advancement and the market offers applications (apps) covering almost every aspect of human life, accessible anywhere and anytime; these have engaged practitioners, providing handy tools for information access, monitoring, telemedicine and even supporting medical decisions.”

Comment 2: provide references to this statement in the discussion ”Technology for mobile devices and AI is rapidly advancing; there is an urgent demand, justified and led by the prevailing patients’ needs, to combine the two fields for a controlled development of quality apps, taken the weaknesses reported so far into account”

Comment 3: Can you include in the discussion any study that have used AI and what were the outcome.

Comment 4: can you define Al vs ANN? As this would help the readers to understand the difference.

Comment 5: in the conclusion, you mentioned the need for further integration of ANN, do you mean AI as well?

Author Response

Reviewer 1

Comment 1: The manuscript does not provide any contribution, evaluation or scientific analysis. It is just a quick discussion about the context and the state of the art. From my point of view, at the current state the paper is not acceptable and even evaluable.

Authors’ actions: We hope that the revised manuscript has addressed all these issues.

Reviewer 3 Report

The authors have presented a thoughtful commentary on limitations of mobile apps, particularly those using deep learning.

I agree with the authors' call for increased accountability and external validation of apps that might be used by consumers to make important health decisions.

A figure that visually supports the authors' proposal would be helpful.

Line 42: Please rephrase this sentence. "Prior to launch" makes more sense than "Considering launch."

Line 45-46: What is meant by Health Insurance accountability? Are you referring to patient privacy?

Author Response

Reviewer 2

Comment 1: provide references to this statement in the introduction “Smartphone utilities demonstrate rapid advancement and the market offers applications (apps) covering almost every aspect of human life, accessible anywhere and anytime; these have engaged practitioners, providing handy tools for information access, monitoring, telemedicine and even supporting medical decisions.”

Authors’ actions: We thank the reviewer for the comment; references added.

Comment 2: provide references to this statement in the discussion ”Technology for mobile devices and AI is rapidly advancing; there is an urgent demand, justified and led by the prevailing patients’ needs, to combine the two fields for a controlled development of quality apps, taken the weaknesses reported so far into account”

Authors’ actions: This statement is now supported by relevant bibliography.

Comment 3: Can you include in the discussion any study that have used AI and what were the outcome.

Authors’ actions: We thank the reviewer for the comment; apart from the references in the third paragraph of the discussion (31-43), we have added a paragraph concerning the use of AI, and the relevant results, linked to one of our previous publications [8].

Comment 4: can you define Al vs ANN? As this would help the readers to understand the difference.

Authors’ actions: In the revised manuscript it is clarified that ANNs is an enabling technology for AI (see revised manuscript with tracked changes in the introduction lines 42-48.

Comment 5: in the conclusion, you mentioned the need for further integration of ANN, do you mean AI as well?

Authors’ actions: Yes we mean AI, the manuscript is corrected, thanks for pointing.

Reviewer 3

Comment 1: A figure that visually supports the authors' proposal would be helpful.

Authors’ actions: The revised manuscript is enhanced with a schematic diagram depicting the concept.

Comment 2: Line 42: Please rephrase this sentence. "Prior to launch" makes more sense than "Considering launch."

Authors’ actions: Corrected as proposed by the reviewer.

Comment 3: Line 45-46: What is meant by Health Insurance accountability? Are you referring to patient privacy?

Authors’ actions: We thank the reviewer for the comment. It now reads: “while Health Insurance accountability should be always considered, in terms of privacy of health information, security of electronic records, administrative simplification, and insurance portability.”.

Round 2

Reviewer 1 Report

Also considering the improvements after the first round of review the paper is not acceptable for publication in the current shape. My main concerns are the following: 

The scientific contribution of the work is absolutely not clear How the proposed steps improve the state of the art ?  How the proposed steps are (or can be) evaluated ?  Comparison with the state of the art should be improved  No implementation information are provided and for the reader is really hard to understand if and how the designed system can be implemented